# Assessment of the Feasibility of Converting the Liquid Fraction Separated from Fruit and Vegetable Waste in a UASB Digester

**DOI:** 10.3390/bioengineering11010006

**Published:** 2023-12-21

**Authors:** Fabrice Tanguay-Rioux, Laurent Spreutels, Caroline Roy, Jean-Claude Frigon

**Affiliations:** Energy, Mining and Environment Research Centre, National Research Council Canada, 6100 Royalmount Ave., Montreal, QC H4P 2R2, Canada; fabrice.tanguay-rioux@cnrc-nrc.gc.ca (F.T.-R.); jean-claude.frigon@cnrc-nrc.gc.ca (J.-C.F.)

**Keywords:** anaerobic digestion, screw press, food waste, UASB, methane

## Abstract

Anaerobic digestion of food waste still faces important challenges despite its world-wide application. An important fraction of food waste is composed of organic material having a low hydrolysis rate and which is often not degraded in digesters. The addition of this less hydrolysable fraction into anaerobic digesters requires a longer hydraulic residence time, and therefore leads to oversizing of the digesters. To overcome this problem, the conversion of the highly biodegradable liquid fraction from fruit and vegetable waste in a up-flow anaerobic sludge blanket (UASB) digester is proposed and demonstrated. The more easily biodegradable fraction of the waste is concentrated in the liquid phase using a 2-stage screw press separation. Then, this liquid fraction is digested in a 3.5 L UASB digester at a high organic loading rate. A good and stable performance was observed up to an organic loading rate (OLR) of 12 g COD/(Lrx.d), with a specific methane production of 2.6 L CH_4_/(Lrx.d) and a degradation of 85% of the initial total COD. Compared to the conversion of the same initial waste with a continuously stirred tank reactor (CSTR), this new treatment strategy leads to 10% lower COD degradation, but can produce the same amount of methane with a digester that is twice as small. The scale-up of this process could contribute to reduced costs related to the anaerobic digestion of food waste, while reducing management efforts associated with digestate handling and increasing process stability at high organic loading rates.

## 1. Introduction

Worldwide, there is over 2 billion tons of municipal solid waste (MSW) generated every year, of which more than 50% corresponds to organic material [1]. These organic residues, often referred to as the organic fraction of municipal solid waste (OFMSW), include several fractions such as source-separated food waste, commercial waste, yard waste, etc. Still today, a large fraction of these waste streams is disposed in landfills, which is associated with large GHG emissions and with an important loss of resources. Recently, significant efforts have been made to divert the OFMSW from landfills by converting these residues into valuable products such as compost or biomethane. One of the main challenging component of OFMSW is food waste since 25–30% of the food produced worldwide is lost and ends up as waste [2]. Food waste alone accounts for approximately 50% of the emissions associated with MSW [1]. Among food waste, fruit and vegetable waste (FVW) represent the main fraction, as it accounts for almost 50% of the total stream [3].

In the last decades, the anaerobic digestion (AD) of food waste has gained considerable interest as a recovery route since it can be used for the treatment of a large variety of organic waste and can produce renewable energy in the form of biomethane [4,5,6,7]. AD is generally seen as a more environmentally friendly technology for food waste valorization compared to other conventional technologies when the complete life cycle is considered [4,8,9]. However, even though AD is a mature technology and is widely used for the treatment of diverse organic waste streams, specific characteristics of FW slow down AD’s full deployment for FW processing due to unresolved challenges [7,10]. One of the main problems is the difficulty of maintaining stable conditions in the digester due to the production and accumulation of volatile fatty acids (VFAs) from the rapid degradation of food waste, leading to the need to maintain a low organic loading rate (OLR) [4,7,8,11,12], and consequently, to have large digester volumes. The conversion of FW by AD can also be inhibited by the accumulation of total ammoniacal nitrogen (TAN) in the digester [13].

This is even more important for fruit and vegetable waste (FVW), as it contains a large fraction of simple sugars that are quickly hydrolyzed during AD, leading to an even more significant production of VFAs and fast acidification of the digester at high OLR [12,14,15,16,17,18]. In addition, FVW generally has a lower biochemical methane potential (BMP) compared to other types of FW [7]. These issues lead to an overall low profitability of the conversion.

For both FW and FVW, solutions have been developed to overcome some of these limitations, such as co-digestion, adequate choice of digester (type and sequence), multistage treatment, and pre-treatment of FW (biological, mechanical, and thermal) [4,6,7,10,16,19]. The direct addition of micro-nutrients to prevent failure from high VFAs or ammonia concentrations has also been identified as a promising strategy [20]. For example, the addition of iron oxide was shown to enhance the anaerobic digestion of food waste [21]. However, none of these strategies have been demonstrated to completely mitigate the limitations faced by the AD of FW, and more research is needed to improve the profitability of the process [7]. For example, the large-scale application of pre-treatment strategies is generally limited by the important additional costs required [19].

In recent years, most studies have aimed at treating the whole FW stream, with or without pre-treatment. However, the presence of solids in the feed is responsible for slowing down the hydrolysis step of AD and preventing the use of high-efficiency digesters [22]. To overcome this limitation, some studies have reported the possibility of recovering soluble compounds from FW in a liquid phase having a high BMP after a separation and/or solubilization step [23,24]. This recovery of the soluble compounds from food waste in a liquid phase could allow for its treatment in a high-efficiency anaerobic digester [24]. Several strategies can be employed, including the extraction of soluble compounds in water via centrifugation [23], the use of a percolation bed [24], and the use of a mechanical press [25,26,27,28]. The extraction of the soluble compounds with a press instead of a percolation method might increase the chemical oxygen demand (COD) recovered, and thus the energetic yield of the process [24]. The use of a screw press separator was also shown to be a more suitable strategy compared to other pre-treatment equipment to remove the less biodegradable organic fraction from food waste, as well as contaminants like plastic and paper [26]. Previous results on the optimization of screw press separation of FVW showed that it is possible to recover up to 65 wt% of food waste in the liquid fraction [25]. Moreover, the solid fraction appears to be easier to handle for composting, or even to be disposed of (less leachate, etc.). Few studies have been carried out on the conversion of the liquid fraction separated from FW to biomethane via AD. The conversion of the liquid fraction in a high-rate anaerobic digester, such as an up-flow anaerobic sludge blanket (UASB), could significantly reduce the digester volume required for the conversion of the liquid fraction due to a higher loading rate, as well as decrease sludge production [22,29]. The advantage of a UASB digester relies in the possibility of decoupling the liquid and the solid fractions’ retention times [29]. Micolucci et al. [30] used a screw press to separate the liquid fraction from the food waste for its conversion in AD, but they used a continuously stirred tank reactor (CSTR) as a digester. Nayono et al. [27] used a semi-continuous UASB digester to convert pressed liquid from food waste and showed that it is possible to reach a stable operation at high OLR (up to an OLR of 27.7 g COD/(Lrx.d)), but with a relatively low COD conversion. However, the pressing process used in their study led primarily to the production of a solid phase, thus leading to low overall methane production. More recently, Orduña-Gaytán et al. [31] converted the liquid fraction from FVW in a fed-batchdigester with an OLR of 44 g COD/(Lrx.d), but only reached a COD conversion of 55%. Gao et al. [32] tested different mixtures of blackwater and food waste at different OLRs in a UASB digester and observed a decrease in the volumetric methane production at an OLR higher than 10 g COD/(Lrx.d). Tsui et al. [33] studied the effect of adding conductive materials to a UASB digester for the two-phase conversion of food waste. However, they worked with synthetic food waste and only operated the digester at relatively low OLR (<5 g COD/(Lrx.d)).

Therefore, the conversion of the liquid fraction from food waste in a UASB digester could reach a similar performance compared to a CSTR, but at a higher loading rate and thus using a smaller digester. The technical comparison of this approach for FW or FVW with a conventional conversion in a CSTR, however, is lacking.

The objective of this study is to validate the performance and stability of processing the liquid fraction separated from FVW with a screw press in a continuous UASB digester and to compare the performance of this treatment strategy to processing the whole FVW stream without a separation pre-treatment in a conventional CSTR anaerobic digester.

## 2. Materials and Methods

FVW was collected and converted through two different approaches. First, the liquid fraction from the waste was separated with a screw press and converted at different organic loading rates (OLRs) in a UASB digester to validate the feasibility of using such an approach for food waste treatment. Then, this strategy was compared to a conventional direct conversion of the same unprocessed FVW in a CSTR digester.

### 2.1. Fruit and Vegetable Waste

The FVW used in this study consisted of a synthetic mix of 6 whole fruits and vegetables mixed with a fixed composition. The fruits and vegetables were chosen arbitrarily to represent a conventional waste stream. Apples, carrots, cucumbers, mandarins, potatoes, and peppers were mixed together in equal mass proportions without pre-processing.

The ingredients were received in bulk from the food bank Moisson Montreal, located in Canada. The resulting FVW stream had a total solid (TS) content and a total volatile solid (TVS) content of 110.1 and 102.7 mg/g, respectively. It had a total chemical oxygen demand (tCOD) of 132.1 g/L. This is similar to values reported in the literature for FVW waste [15,34]. The TVS/TS ratio was high (93%), indicating a high organic content, which is important for AD [35].

### 2.2. Screw Press Pre-Treatment

The FVW was pre-processed with a screw press in about 20 batches. The composition of the waste was kept constant in terms of the ingredients and their relative proportions. The screw press pre-processing consisted of 3 main steps: the first screw press separation (SP1), where the entire waste stream is treated in the screw press, thus producing a liquid fraction (LIQ1) and a solid fraction (SOL1); the maceration of the solid, where SOL1 resulting from SP1 is mixed with water in a mass ratio of 1:1 and set aside at room temperature for 24 h; and the second screw press separation (SP2), where the macerated mixture is treated in the screw press, thus producing a liquid fraction (LIQ2) and solid fraction (SOL2). The liquids produced from both screw press separation steps (LIQ1 and LIQ2) were mixed together to be used as substrate during anaerobic digestion, while the residual solid fraction (SOL2) was set aside for this project. This fraction could potentially be used as substrate for compost production or further converted in a thermochemical process for energy valorization.

For both screw press separation steps, a CP-4 model (Vincent Corporation, Tampa, FL, USA) was used, and it was operated in both cases at 20 rpm with an applied pressure of 60 psig according to previously optimized conditions [25].

### 2.3. Inoculum

The same inoculum was used for both digesters. It consisted of granular biomass (mixed population) collected from a full-scale UASB digester processing industrial wastewater (Lassonde Inc., Rougemont, QC, Canada). The drained granulated biomass had a TS and a TVS of 90 and 79 mg/g, respectively.

### 2.4. Up-Flow Anaerobic Sludge Blanket Digester

The UASB used is a cylindrical glass digester having a diameter and a height of 0.06 m and 1.15 m, respectively. The volume of liquid in the digester is 3.4 L, while the volume of the headspace is 0.93 L. The liquid in the digester was recirculated at a rate of 0.095 L min^−1^, leading to an up-flow velocity of 2 m/h. The digester was maintained at 35 °C using a water jacket. The pH was maintained at 7.0 ± 0.2 by adding NaOH 2 N.

On start-up, 1.1 kg of drained granulated biomass was added to the digester, and the rest of the volume was filled with 2.3 L of 1 M bicarbonate buffer.

The digester was operated continuously for almost 300 d. It was fed continuously by adding the liquid fraction of the FVW every 30–60 min. The feed consisted of the mix of LIQ1 and LIQ2 and was prepared as described in Section 2.2. It was also diluted with water to reach a tCOD of 25 g/L. In addition to the liquid fraction of FVW fed to the digester, varying quantities of bicarbonate buffer and defined media, as detailed in [36], were fed every day to account for 4% and 6% of the feed, respectively.

The OLR was gradually increased from 2 to 52 g COD/(Lrx.d) in order to test the stability and the limits of the system. In the first phases, the OLR was gradually increased once a new steady-state was reached and kept constant for a few days. After reaching an OLR of 21 g COD/(Lrx.d), the OLR was rapidly increased to observe the effect on the stability of the digester.

### 2.5. Continuously Stirred Tank Digester

The CSTR used is a cylindrical plastic digester having a diameter and a height of 0.254 m and 0.318 m, respectively. The volume of liquid in the digester is 10 L, while the volume of the headspace is 5 L. The digester was continuously agitated with an impeller at 50 rpm. The digester was maintained at 35 °C using a water jacket. The pH was maintained at 7.0 ± 0.2 by adding NaOH 2 N.

On start-up, 3.5 kg of drained granulated biomass, 2 L of 1 M bicarbonate buffer, and 2 L of water were added to the digester, and the rest of the volume was filled with substrate.

The CSTR was operated continuously for 170 d. It was fed once every weekday by pumping in the mixture of FVW directly without pre-treatment. In addition to the FVW fed to the digester, a varying quantity of the defined medium was fed every day to account for 4.5% of the feed.

The digester was started with an OLR of 2 g COD/(Lrx.d), and it was gradually increased up to an OLR of 6 g COD/(Lrx.d) to test the limit of the system. The OLR was increased after operating for a few days at steady-state.

### 2.6. Analytical Measurements

The total solids (TS), the total volatile solids (TVS), the suspended solids (SS), and the volatile suspended solids (VSS) content were characterized according to standard methods (APHA, 2005). Soluble and total chemical oxygen demand (sCOD and tCOD) were also measured according to standard methods (APHA, 2005). The pH was measured using a Thermo Scientific Orion Star pH meter. Volatile fatty acids (VFAs), including acetic acid, propionic acid, butyric acid, valeric acid, and caproic acid were quantified via GC-FID using an Agilent 7890B gas chromatograph (Santa Clara, CA, USA) equipped with a Nukol capillary column (30 m × 0.25 mm × 0.25 µm). Cations (Na^+^, NH_4_^+^, K^+^) were measured by injecting a sample volume of 20 µL into a high-performance liquid chromatograph (HPLC) (Waters Chromatography, Milford, MA, USA) equipped with a Hamilton PRP-X200 cation resin-based column (250 × 41 mm O.D.), a conductivity detector (Waters Millipore model 432), and an Empower2 data station.

Measurements of H_2_, N_2_, O_2_, CO, CH_4_, and CO_2_ were made on an Agilent 7820 gas chromatograph (Wilmington, DE, USA) coupled to a thermal conductivity detector (TCD). The gas sample was injected on a 2 m × 2 mm I.D. ShinCarbon ST packed column from RESTEK. Argon was used as carrier gas at a flow rate of 10 mL min^−1^.

## 3. Results and Discussion

### 3.1. Screw Press Separation of Fruit and Vegetable Waste

The processing of the FVW with the two successive screw press separation steps led to the recovery of 68% of the initial TS and TVS, and to the recovery of 72% of the tCOD in the liquid phase (LIQ1 + LIQ2). The repartition of the total mass, the TS, the TVS, and the tCOD for each step of the separation process is presented in Table 1. For the calculation, the loss of material occurring during the screw press separation was neglected since it is proportional to the volume of the equipment rather than to the quantity of waste processed. Therefore, processing large quantities of waste in a continuous industrial application would lead to negligible loss of material.

Despite recovering 67% of the initial mass of the FVW in LIQ1 during SP1, only 50% of the TS and TVS, and 54% of the tCOD were recovered in this fraction. However, as determined during the screw press process optimization [25], the addition of the maturation step followed by a second screw press separation led to a significant increase in the TS, TVS, and tCOD recovered in the global liquid phase, as 68%, 68%, and 72% of the TS, TVS and tCOD were recovered, respectively. Globally, a smaller fraction of organic compounds was recovered in the liquid phase during SP2 compared to SP1 since most of these compounds were already separated. Still, the addition of the second screw press separation increases the recovery of tCOD by 25%, demonstrating the importance of this second step. The overall recovery of 72% of the tCOD in the liquid fraction represents the maximal theoretical conversion that could be achieved through the combined screw press-UASB process in comparison to a treatment of the entire stream with a CSTR. However, the real difference between these two approaches for COD conversion is expected to be lower since most of the organic compounds that have a slow hydrolysis rate or that are non-biodegradable are expected to end up in the solid fraction. Globally, the recovery of the tCOD obtained with the screw press is significantly higher than the ones reported with other separation methods such as direct extraction with water [23] and extraction with a percolation bed [24].

Due to the addition of water during the maceration step, 1.12 kg of liquid product (LIQ1 + LIQ2) was obtained for 1 kg of FVW. In this case, 0.33 kg of water was added. The liquid obtained had a TS, TVS, SS, and VSS of 69.6 mg/g, 64.6 mg/g, 27.4 mg/g, and 26.4 mg/g, respectively. It also had a tCOD of 93.6g/L and a sCOD of 66.1 g/L, leading to a sCOD/tCOD ratio of 0.71.

### 3.2. Conversion of FVW with a UASB Digester

The entire operation of the UASB digester was separated into 16 phases according to the OLR and the performance of the system. The main performance indicators and operating conditions for these phases are presented in Table 2. The volumetric methane production and the methane yield were calculated by considering the entire production of CH_4_ for the phase, while the concentration of VFAs and the COD conversion were calculated as averages of punctual measurements (approximately every week) taken during the different phases. Standard deviations are provided for the variables that were calculated based on averages (i.e., concentration of VFAs and COD conversion).

From phases 1 to 10, the OLR was gradually increased from 2 to 12 g COD/(Lrx.d). However, operational problems occurred during phases 2 and 7, leading to the need to decrease the OLR and wait for the stabilization of the digester before increasing it back again. During phase 2, a problem occurred with the gas counter, hence the absence of measurements for the gas production in Table 2. At the beginning of phase 7, there was a blockage in the recirculation and the effluent lines. Part of the liquid in the digester was removed and fresh biomass was added. The OLR was thus decreased momentarily to wait for the new biomass to adapt. Apart from these 2 phases, during the 10 first phases, the volumetric methane production increased gradually with the OLR, leading to a relatively high methane yield (>0.22 L CH_4_/g COD Fed) and low concentrations of VFAs (<240 mg/L). At an OLR of 12 g COD/(Lrx.d) (phase 10), the COD conversion decreased from more than 0.91 to 0.85, probably due to the decrease in the hydraulic retention time (HRT). The further increase in the OLR to 21 g COD/(Lrx.d) in phase 11 led to another important decrease in the COD conversion and methane yield (decrease of 23% in the methane yield from phase 10 to 11). Therefore, the highest OLR that was reached without decreasing the methane yield was 12 g COD/(Lrx.d) (HRT of 2.1 d), while the highest OLR that was reached without significantly decreasing the COD conversion was 10 g COD/(Lrx.d) (HRT of 2.6 d). After phase 11, the OLR was further increased rapidly (phases of 2–7 d) to test the limit of the system. These phases were shorter due to the technical challenges of feeding a digester with such a high OLR in laboratory conditions, but as the HRT was also shorter, the impact of the changes in OLR were quickly observable. This increase led to a gradual decrease in performance up to an OLR of 50 g COD/(Lrx.d), at which point the gas production was too high, leading to a washout of the biomass. Finally, the OLR was decreased back to 28 g COD/(Lrx.d) to verify if the digester could recover. Interestingly, it was possible to come back to a similar performance as before, as no accumulation of VFAs or other inhibitors was detected. UASB digesters are known to be able to handle organic overloads adequately [29], which was demonstrated here with food waste.

Globally, the UASB digester was operated for almost 300 d without major operational failure, thus demonstrating the feasibility of converting the liquid fraction separated from FVW in a UASB digester. During the entire operation, the methane concentration in the biogas remained stable at 53 ± 4%, and pH was kept constant between 6.9 and 7.2. Figure 1 shows the volumetric methane production, the methane yield, and the tCOD of the feedstock and the effluent. As seen in Figure 1C, the tCOD of the feedstock was stable for the entire operation with an average concentration of 25.2 ± 2.4 g/L. However, while a good degradation of the tCOD was observed for the first 200 d (OLR < 10) with an average value of 1.6 ± 1.1 g/L for the residual tCOD, higher values were measured when the OLR was further increased. This is perfectly normal and correlated with the biomass production at higher OLR. When the OLR was above 10, an average residual tCOD in the effluent of 4.8 ± 2.1 g/L was measured. However, despite the increase in the tCOD in the effluent, this was not observed for the sCOD, as a concentration of 0.53 ± 0.25 g/L was measured for the entire operation. This shows that despite the increase in the OLR, the soluble organic compounds were still almost completely converted and the residual COD in the effluent is due to the slower degradation of non-soluble organic compounds at higher OLR and to the washout of biomass growth from the digester.

The variation in the volumetric methane production and the methane yield as a function of the OLR can be seen in Figure 2A. The volumetric methane production increased linearly with the OLR until an OLR of 12 g COD/(Lrx.d), after which it increased more slowly until reaching its maximum due to physical limitations related to the experimental setup. As for the methane yield, it remained relatively constant until the same OLR value, after which it decreased.

A similar trend was observed for the COD conversion (Figure 2B) as it decreased with the OLR until reaching a relatively stable conversion of about 0.75 for an OLR between 20 and 40 g COD/(Lrx.d). For the higher OLR, the standard deviation of the COD conversion was higher, leading to less precision in the prediction. Moreover, Nayono et al. [27] also observed a gradual decrease followed by a plateau for the COD conversion for a similar setup and substrate, but they observed a lower conversion (60–75% for an OLR higher than 11 g COD/(Lrx.d)), probably due to a more concentrated initial substrate. At a high OLR, the COD conversion observed is also higher than the one observed for the conversion of the liquid fraction of FVW by [31], probably due to the use of a different type of digester.

Different operating parameters were measured during the operation of the digester to ensure its good performance, including the concentration of VFAs and the main cations concentration. The main VFAs measured in the effluent were acetate, propionate, and butyrate with respective concentrations of 75 ± 18%, 17 ± 14%, and 8 ± 14%. During the operation, the repartition of these three organic acids changed, but it was not possible to correlate this change to any other operating parameter. As seen in Table 1, the concentration of VFAs remained below 100 mg/L for the first 200 d (OLR < 10). Then, it increased with the OLR, but never reached an average concentration higher than 600 mg/L, which is lower than the values reported for inhibition from VFAs [11]. Generally, the trend of the concentration of VFAs was similar to the one of the tCOD of the effluent. As for the cations, no accumulation was observed at a higher OLR. The concentration of ammonium was at its highest during phase 6 with a value of 1140 ± 150 mg/L, which is slightly higher than the TAN concentration of 1000 mg/L at which [13] observed a decrease of 10% in the methane production in a single-stage digester for food waste conversion. However, this inhibition was reported for a CSTR operation and it is not clear if it applies also to a UASB digester. Also, for a higher OLR (more than 20 g COD/(Lrx.d)), the ammonium concentration decreased around 600–700 mg/L, which corresponds to more reasonable concentrations. Still, the effect of the high ammonium concentration would need to be further investigated for future optimization of the system.

### 3.3. Conversion of FVW with a CSTR Anaerobic Digester

The CSTR was used to directly convert the raw FVW into biomethane. The OLR was gradually increased over a period of 175 d. The entire operation of the CSTR digester was separated into five phases according to the OLR and the performance of the system. The main performance indicators and operating conditions for these phases are presented in Table 3. The volumetric methane production and the methane yield were calculated, as for the UASB, by considering the entire production of CH_4_ for the phase. The concentration of VFAs, the COD conversion, and the pH were calculated as the average of punctual measurements taken during the different phases.

The CSTR was started with an OLR of about 1.5, which was then increased to 2.5 after 20 d (Phases 1 and 2). After 80 d, the digester reached a steady-state, and the OLR was increased to 3 for 35 d (Phase 3). Finally, the OLR was increased to 4 and 6 g COD/(Lrx.d) for 28 d each (Phases 4 and 5). During phase 4 (OLR of 4 g COD/(Lrx.d)), the highest methane yield was achieved, which is consistent with the literature for the conversion of FVW in a CSTR [15]. At this point, the averages of volumetric methane production and methane yield reached 1.2 L CH_4_/(Lrx.d) and 0.27L CH_4_/COD fed, respectively, for the entire phase. The methane yield obtained in Phase 4, corresponding to about 0.4 L CH_4_/g TVS fed, is similar to that reported in literature for FVW conversion in a single-stage CSTR [34]. At a higher OLR (OLR of 6), the concentration of VFAs gradually started to increase in the digester and the methane yield decreased. This can be seen in Figure 3, which presents the characteristics of the effluent of the CSTR over time. At the end of Phase 5, the digester was stopped due to this gradual accumulation of VFAs and the decrease in the performance. Overall, the COD conversion remained approximately constant at 75% for all the phases, except at the end. The methane concentration remained stable at 54 ± 5%, while the pH was kept constant between 7 and 7.2.

Results show that the sCOD of the effluent started to increase in the last phase, demonstrating a gradual decrease in the performance of the digester. The sCOD increased from 2.6 ± 0.3 g/L in Phase 4 to 3.8 ± 0.8 g/L in the first half of Phase 5, and eventually to 6.2 ± 3.2 g/L in the second half of Phase 5. At the end of the operation, the sCOD reached 12 g/L. As for the VFAs, Figure 3 clearly shows a rapid accumulation of both acetate and propionate in the last days of operation, while their concentration remained low for the four first phases, with an average total concentration of VFAs of 200 mg/L. In the last days before stopping the digester, the concentration of VFAs reached 5 g/L, mainly as propionate, showing an important accumulation. According to the substrate, inhibition of AD from VFAs was reported to occur at concentrations higher than 2–4 g/L [37]. Again, this accumulation of VFAs, mainly under the form of propionate, is in accordance with the literature for the conversion of FVW in a single-phase CSTR at high OLR [12,15].

### 3.4. Comparison of the Strategies for Converting Fruit and Vegetable Waste into Biomethane

Two strategies were used to convert FVW into biomethane. In the first strategy, the waste was separated using a screw press pre-treatment procedure into a liquid and a solid fraction. The liquid fraction was converted with a UASB digester at varying OLRs, while the solid fraction, which has a lower moisture content and is composed mainly of insoluble compounds, was not used in this study. In the second strategy, which corresponds to the traditional approach, the entire FVW stream was directly converted in a CSTR at varying OLRs without pre-treatment. Both digesters (UASB and CSTR) were operated for a long period at varying OLRs in order to identify their optimal operating conditions. In both cases, the OLR was increased until reaching a maximum value, which is why they were not operated at the same OLR and for the same time period.

For both conversion strategies, a maximum methane yield of about 0.26–0.27 L CH_4_/g COD fed was achieved at low OLR (3–4 g COD/(Lrx.d)). At such an OLR, the volumetric methane production was slightly higher for the CSTR, while the COD conversion was significantly higher for the UASB digester (0.93 for the UASB vs. 0.76 for the CSTR). However, when considering the global COD conversion of the entire process, thus including the removal of organic material with the screw press (solid fraction), the COD conversion of the screw press-UASB process was 0.67 for an OLR of 4 g COD/(Lrx.d). This is smaller than the global COD conversion obtained for the CSTR. Nevertheless, it was possible to operate the UASB digester at significantly higher OLRs compared to the CSTR, thus leading to lower HRTs. At an OLR of 3–4 g COD/(Lrx.d) (maximum methane yield), the HRT of the CSTR was 21–27 d, while it was only 7 d for the UASB. Furthermore, the lowest HRT achieved in the CSTR was 16 d, while a stable operation was achieved in the UASB at an HRT lower than 1 d. In addition, the CSTR showed signs of VFA accumulation at an OLR of 6 g COD/(Lrx.d), while no sign of VFA accumulation was observed for the UASB, even at a very high OLR.

Figure 4 shows the performance of the entire screw press-UASB system according to its OLR in comparison to the optimal condition of the CSTR (OLR of 4 g COD/(Lrx.d)) for the conversion of FVW. Depending on the OLR, the screw press-UASB system can convert 93 to 68% of the tCOD that would be converted in a CSTR (left axis of Figure 4), but with a digester volume 0.3 to 6 times smaller for the same CH_4_ production (right axis of Figure 4). Therefore, as the OLR of the UASB digester increases, a smaller portion of the tCOD fed is converted in comparison to the CSTR, but a lower digester volume is required for the same CH_4_ production. In theory, a UASB digester six times smaller than a conventional CSTR could be used with this new treatment strategy at an OLR of about 40 g COD/(Lrx.d), but at the cost of globally converting only 70–75% of the tCOD that would be converted in a CSTR.

Based on the results obtained in this study, a good compromise between these two parameters is obtained with an OLR of 10 g COD/(Lrx.d), as 90% of the conversion of the tCOD is achieved, but with a digester twice smaller than a conventional CSTR operated in optimal conditions. This choice of OLR is, however, very conservative, as results indicate that it would be possible to reach a higher OLR for the UASB, and thus further decrease the volume of the digester needed. In addition to reducing the digester size, the combination of a screw press and a UASB digester for the conversion of FVW can also increase the stability of such conversion, as it was shown that despite increasing the OLR to considerably high values, no inhibition nor accumulation of VFAs occurred, and the digester was able to recover. Finally, this strategy could also reduce the challenges associated with digestate handling, as no digestate is produced from the UASB digester and the solid fraction separated from the screw press could be easier to convert into compost than digestate. Indeed, this solid fraction has a lower moisture content than digestate, which facilitates its conversion into compost by reducing the need for the addition of structural material [27].

This work demonstrated the feasibility and the interest of using a screw press and a UASB digester to convert FVW. However, the configuration and the operating conditions were not optimized. Further work should focus on improving the COD conversion and the volumetric methane production of the waste in a UASB at high OLR. Potential improvements could further increase the techno-economic viability of such process, which would need to be demonstrated in the future based on a techno-economic assessment. This optimization could be done through different strategies, including, for example, by increasing the recovery of organic compounds in the liquid fraction during the screw press, improving the UASB digester design, increasing the initial COD of the feed, adding a pre-hydrolysis step before the UASB [22] and enhancing biomass granulation with chemical additives [38]. In the future, the implementation of novel in-line monitoring and control strategies to maximize the OLR without perturbating the microbial community could also potentially improve the performance of the process [39]. The effect of the TAN concentration on the conversion of FVW in a UASB digester would also need to be addressed. Finally, future work will also need to evaluate the potential for the conversion of the solid separated from the screw press in order to assess its technical feasibility.

## 4. Conclusions

The conversion of the liquid fraction separated from FVW in a UASB digester at a high OLR was demonstrated, and the performance was compared to direct conversion in a CSTR. Separating FVW into two phases (solid and liquid) with a screw press allows the produced liquid to be converted in a high-efficiency digester, while the produced solid can be more easily composted due to a lower moisture content. A stable performance of the UASB digester was observed up to an OLR of 44 g COD/(Lrx.d), but the COD conversion and the methane yield both decreased significantly from an OLR higher than 10–12 g COD/(Lrx.d). Compared to converting the same initial waste with a CSTR, the combination of a screw press and a UASB digester operating with a conservative OLR of 10 g COD/(Lrx.d) led to a degradation of the COD 10% smaller, but made it possible to produce the same quantity of methane with a digester twice as small.

## Figures and Tables

**Figure 1 bioengineering-11-00006-f001:**
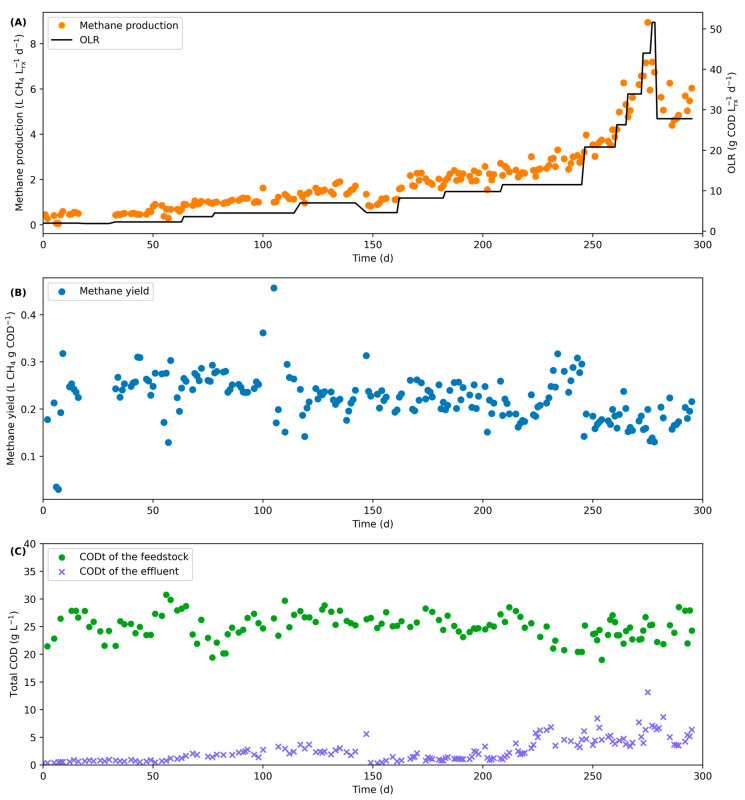
Production of methane (**A**), methane yield (**B**), and tCOD of the feedstock and the effluent (**C**) obtained during the conversion of FVW in the UASB digester with respect to the time of operation and the OLR.

**Figure 2 bioengineering-11-00006-f002:**
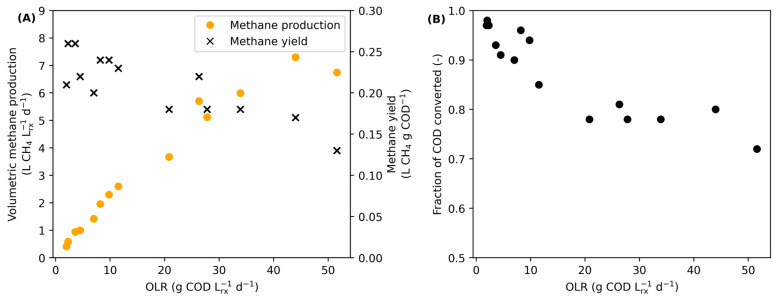
Volumetric methane production and the methane yield according to the OLR (**A**), and fraction of COD converted according to the OLR (**B**) for the UASB digester.

**Figure 3 bioengineering-11-00006-f003:**
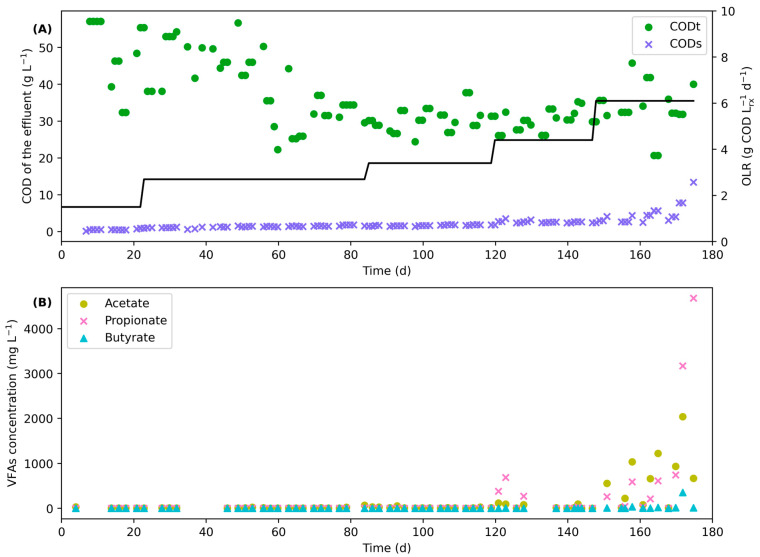
tCOD and sCOD of the effluent (**A**) and concentration of VFAs (**B**) of the effluent in the CSTR anaerobic digester with respect to the time of operation for the conversion of FVW.

**Figure 4 bioengineering-11-00006-f004:**
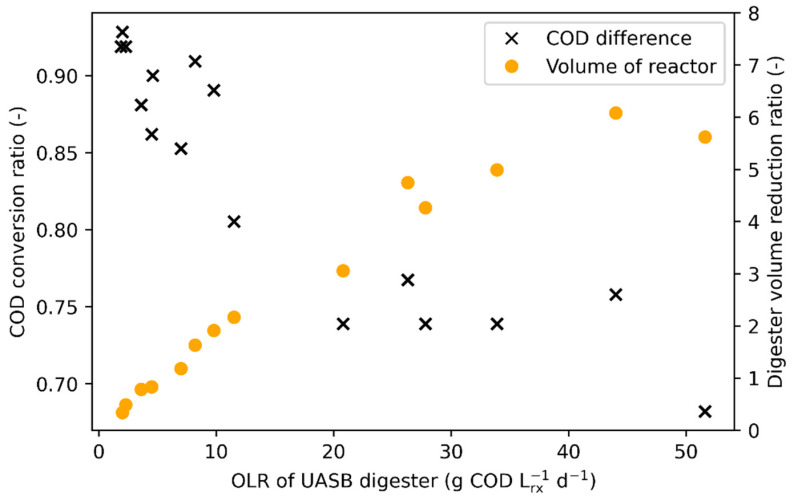
Conversion of FVW with a UASB digester compared to a CSTR digester operated at an OLR of 4 (optimal conditions). Results are provided as a ratio between both digesters for varying OLRs of the UASB digester.

**Table 1 bioengineering-11-00006-t001:** Performance of the screw press separator for the recovery of the organic compounds from FVW (kg/kg of feed).

	Repartition of the Total Mass	Repartition of the TS	Repartition of the TVS	Repartition of the tCOD
Feed for SP1 (FVW)	1	1	1	1
LIQ1	0.67	0.50	0.50	0.54
SOL1	0.33	0.50	0.50	0.46
Feed for SP2	0.66	0.50	0.50	0.46
LIQ2	0.45	0.18	0.18	0.18
SOL2	0.21	0.32	0.32	0.28
LIQ1 + LIQ2	1.12	0.68	0.68	0.72

**Table 2 bioengineering-11-00006-t002:** UASB performance for the conversion of FVW for the different phases of operation.

Phases	Number of Days	OLR (g COD/(Lrx.d))	HRT (d)	COD Conversion (-)	Volumetric Methane Production (L CH_4_/(Lrx.d))	Methane Yield (L CH_4_/g COD Fed)	VFA Concentration (mg/L)
Phase 1	16	2	13.1	0.98 ± 0.00	0.4	0.21	22 ± 21
Phase 2	14	2	13.2	0.97 ± 0.01	-	-	20 ± 20
Phase 3	33	2	11.9	0.97 ± 0.01	0.6	0.26	41 ± 59
Phase 4	14	4	6.9	0.93 ± 0.01	0.9	0.26	49 ± 13
Phase 5	37	5	5.8	0.91 ± 0.02	1.0	0.22	73 ± 69
Phase 6	28	7	3.9	0.90 ± 0.02	1.4	0.20	87 ± 128
Phase 7	19	5	5.8	0.95 ± 0.07	0.8	0.17	11 ± 4
Phase 8	21	8	3.2	0.96 ± 0.02	2.0	0.24	50 ± 46
Phase 9	26	10	2.6	0.94 ± 0.03	2.3	0.24	94 ± 60
Phase 10	37	12	2.1	0.85 ± 0.07	2.6	0.23	237 ± 105
Phase 11	15	21	1.2	0.78 ± 0.07	3.7	0.18	246 ± 72
Phase 12	5	26	0.9	0.81 ± 0.03	5.7	0.22	151 ± 83
Phase 13	7	34	0.7	0.78 ± 0.09	6.0	0.18	188 ± 8
Phase 14	4	44	0.6	0.8 ± 0.05	7.3	0.17	569 ± 70
Phase 15	2	52	0.5	0.72	6.8	0.13	-
Phase 16	17	28	0.9	0.78 ± 0.09	5.1	0.18	336 ± 200

**Table 3 bioengineering-11-00006-t003:** CSTR performance for the conversion of FVW for the different phases of operation.

Phases	Number of Days	OLR (g COD/(Lrx.d))	HRT	COD Conversion (-)	Volumetric Methane Production (L CH_4_/(Lrx.d))	Methane Yield (L CH_4_/g COD)	VFA Concentration (mg/L)
Phase 1	22	1.5	71	0.69 ± 0.06	0.3	0.21	11 ± 12
Phase 2	62	2.5	40	0.77 ± 0.04	0.5	0.19	14 ± 15
Phase 3	35	3	27	0.75 ± 0.03	0.9	0.26	16 ± 15
Phase 4	28	4	21	0.76 ± 0.02	1.2	0.27	200 ± 280
Phase 5	28	6	16	0.75 ± 0.05	1.5	0.24	2023 ± 2000

## Data Availability

Data are contained within the article.

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
