# Peer review of "Assessment of the Feasibility of Converting the Liquid Fraction Separated from Fruit and Vegetable Waste in a UASB Digester"

_bioengineering, 2023, doi:10.3390/bioengineering11010006_

Round 1

Reviewer 1 Report

Comments and Suggestions for Authors

Some diferences between the comparison UASB/CSTR are highlighted: duration (300/170 days), size of digester, inoculum ratio, OLR... Does this affect the results? something about must be said

According to figure 1, the higher methane production appears from day 250. This is not too yield.

Find more comments in the pdf

Author Response

Comment 1.1: Some diferences between the comparison UASB/CSTR are highlighted: duration (300/170 days), size of digester, inoculum ratio, OLR... Does this affect the results? something about must be said

Answer: Two sentences were added at the end of the first paragraph of section 3.4. These differences do not affect the results since the objective was to identify the optimal operating conditions, which were significantly different for both digesters, thus explaining the differences in the operating conditions.

Comment 1.2: According to figure 1, the higher methane production appears from day 250. This is not too yield.

Answer: The higher methane production was achieved after 250 days because the OLR was increased very gradually in order to monitor the different phases during this study. However, it is expected that for an industrial operation, the OLR could be increased much faster, thus leading to a high methane production faster.

Additional comments from the pdf:

Answer: Most of the comments found in the pdf were directly addressed in the manuscript.

Other comments:

  • rx in Lrx stands for reactor.
  • wt in wt% stands for total weight
  • the 67% and the 68% in section 3.1 refer to 2 different values in the Table.

Reviewer 2 Report

Comments and Suggestions for Authors

The authors have presented their work in a manuscript entitled “Assessment of the feasibility of converting the liquid fraction separated from fruit and vegetable waste in a UASB digester.” The authors have tried to validate the performance and stability of processing the liquid fraction separated from fruit and vegetable waste with a screw press in a continuous up-flow anaerobic sludge blanket (UASB) digester and to compare the performance of this treatment strategy to processing the whole fruit and vegetable waste stream without a separation pretreatment in a CSTR anaerobic digester. However, the manuscript needs minor revisions to be accepted for publishing in the Bioengineering journal. The following are my observations and comments:

1.      Incorporate recent references into the introduction section.

2.      In section 2.3, the name of the inoculum is not specified. This ensures precision and clarity in detailing the experimental procedure.

3.      In Table 1 include the units utilized, to provide readers with a comprehensive understanding of the presented data.

4.      Check and rectify any instances of excessive spacing after punctuation, especially after full stops. Specifically, in line 263, ensure that there is a single space following each full stop to maintain consistency and readability in the document.

5.      Expand the discussion and analysis within the results section by elaborating on findings, implications, and connections to existing literature. Expanding the discussion enriches the depth and insight of the study's outcomes.

6.      The results are not properly discussed with the recent reported studies. I suggest comparing the results of the present study with the previous studies.

Author Response

The authors have presented their work in a manuscript entitled “Assessment of the feasibility of converting the liquid fraction separated from fruit and vegetable waste in a UASB digester.” The authors have tried to validate the performance and stability of processing the liquid fraction separated from fruit and vegetable waste with a screw press in a continuous up-flow anaerobic sludge blanket (UASB) digester and to compare the performance of this treatment strategy to processing the whole fruit and vegetable waste stream without a separation pretreatment in a CSTR anaerobic digester. However, the manuscript needs minor revisions to be accepted for publishing in the Bioengineering journal. The following are my observations and comments:

Comment 2.1: Incorporate recent references into the introduction section.

Answer: A few more recent references have been added to the introduction section and the discussion.

Comment 2.2: In section 2.3, the name of the inoculum is not specified. This ensures precision and clarity in detailing the experimental procedure.

Answer: The inoculum is a mixed population. Information was added in the text.

Comment 2.3: In Table 1 include the units utilized, to provide readers with a comprehensive understanding of the presented data.

Answer: Units were added in the legend of the table.

Comment 2.4: Check and rectify any instances of excessive spacing after punctuation, especially after full stops. Specifically, in line 263, ensure that there is a single space following each full stop to maintain consistency and readability in the document.

Answer: The manuscript was double-checked entirely to remove any occurrence.

Comment 2.5: Expand the discussion and analysis within the results section by elaborating on findings, implications, and connections to existing literature. Expanding the discussion enriches the depth and insight of the study's outcomes.

Answer: The discussion and the analysis were expanded, mainly in section 3.2 and section 3.4.

Comment 2.6: The results are not properly discussed with the recent reported studies. I suggest comparing the results of the present study with the previous studies.

Answer: See answer to comment 2.5.

Reviewer 3 Report

Comments and Suggestions for Authors

Dear Authors,

I read with pleasure your manuscript with an actual and increasing interest subject in the field of waste treatment and valorization, especially considering the organic waste (as  MSW). I like your manuscript but before acceptance there are required certain corrections and additional information inserted. You must consider my comments and recommendations for improvement of your manuscript quality.

Specific comments and recommendations:

1. In the abstract, must be firstly explained what means 'OLR' and after use it.

2. It can be inserted an explanatory glosarry with all defined abbreviations used in this manuscript before the References section.

3. Page 3 line 104 - correction required, i.e. instead of term 'water' use 'liquid' and thus corrected text will be '...pressed liquid...'.

4. Page 3, section 2.1. - information must be inserted about the fruirt/vegetable waste sample tested. It must be mentioned what parts of wasted vegetables and fruits were used (?), e.g. the complete fruits and vegetables or only the peels, rests of pulps for being a representative model for  MSW.

4. Page 3 line 139 - insert before the term 'model' its name, i.e. shortly as follows '... CP-4 model...'. In an other following paragraph, you give the information about the main characteristics of equipment used associated with the producer company.

5. Page 3 line 141 - change the words' order in the sentence such as '...pre-processing screw press...'.

6. Page 4 line 167 - verify the value of NaOH concentration for pH adjustment. It is not too high ? or you were used diluted solutions prepared from the stock one mentioned ?

7. Page 4 line 170 - correct the section number. It is 'section 2.2' instead of 'section 0'. 

8. Page 4 - please verify the mentioned volume of liquid in digester and the volume of headspace. If the volume of liquid in digester is 10 L, with the sizes mentioned in the manuscript, the total calculated volume of digester is 14.72 L and the headspace can be of 5 L ?

9. Page 4 line 188 - correction needed for word 'weekday'.

10. Page 4 line 267 - it must be defined the expression used for calculation of COD conversion at the experimental part.

11. Page 9 line 312 - the title of Figure 2 must be a little bit modified. Separate explanation for (A) and (B) which will follow after '.' and before these instead of 'Variation' can be used a formulation as 'Balance of main compounds produced and consumed. (A)...' or an other introductive formulation.

12. Page 9 line 321 - text correction as 'Different operating parameters...'.

13. Page 10 line 372 - text correction as '...for the first four phases,...'. (not '5' but 'five').

14. Page 11 line 380 - Figure 4 - title correction as '....tCOD and sCOD...(A) ....(B) of the effluent in the CSTR ...'

15. Page 12 - correction required for:

(i) line 414 - delete text 'that will be converted' and the text will remain as 'but at the cost of globally converting only 70-75% of the tCOD in a CSTR.';

(ii) line 421 - correction as 'tCOD in the CSTR is achieved.' (too much 'of');

(iii) line 438 - possible correction of the following text at the fifth word or related to it and the second word -  'Potential im-provements could further improve the techno-economic viability of such process,...'.

16. The authors' contribution must be mentioned.

17. An English spelling check can be performed for verifying no other mistakes presence. 

These are only a few of my comments and recommendations for the manuscript authors.

Manuscript Reviewer

Author Response

Dear Authors,

I read with pleasure your manuscript with an actual and increasing interest subject in the field of waste treatment and valorization, especially considering the organic waste (as  MSW). I like your manuscript but before acceptance there are required certain corrections and additional information inserted. You must consider my comments and recommendations for improvement of your manuscript quality.

Specific comments and recommendations:

Comment 3.1: In the abstract, must be firstly explained what means 'OLR' and after use it.

Answer: The proposed change has been made.

Comment 3.2: It can be inserted an explanatory glosarry with all defined abbreviations used in this manuscript before the References section.

Answer: We don’t believe that a glossary is necessary as only commonly used acronyms were used and they were all defined in the text.

Comment 3.3: Page 3 line 104 - correction required, i.e. instead of term 'water' use 'liquid' and thus corrected text will be '...pressed liquid...'.

Answer: The proposed change has been made.

Comment 3.4: Page 3, section 2.1. - information must be inserted about the fruirt/vegetable waste sample tested. It must be mentioned what parts of wasted vegetables and fruits were used (?), e.g. the complete fruits and vegetables or only the peels, rests of pulps for being a representative model for  MSW.

Answer: It was specified in section 2.1 that the whole fruits and vegetables were used without pre-processing.

Comment 3.5: Page 3 line 139 - insert before the term 'model' its name, i.e. shortly as follows '... CP-4 model...'. In an other following paragraph, you give the information about the main characteristics of equipment used associated with the producer company.

Answer: CP-4 is the model name. There is no other name for this equipment provided by the manufacturer.

Comment 3.6: Page 3 line 141 - change the words' order in the sentence such as '...pre-processing screw press...'.

Answer: We don’t think the formulation should be changed.

Comment 3.7: Page 4 line 167 - verify the value of NaOH concentration for pH adjustment. It is not too high ? or you were used diluted solutions prepared from the stock one mentioned ?

Answer: The concentration of NaOH mentioned in the text was verified and is adequate. It is not too high for pH control as only small quantities were injected with a small peristaltic pump. A high concentration was chosen to prevent the dilution of the reactor content.

Comment 3.8: Page 4 line 170 - correct the section number. It is 'section 2.2' instead of 'section 0'.

Answer: The proposed change has been made.

Comment 3.9: Page 4 - please verify the mentioned volume of liquid in digester and the volume of headspace. If the volume of liquid in digester is 10 L, with the sizes mentioned in the manuscript, the total calculated volume of digester is 14.72 L and the headspace can be of 5 L ?

Answer: The dimension of the reactor mentioned were rounded. The actual volume of the digester is 16L, but there are tubes in the digester. So, the working volume is 15L. Changes were made to the manuscript to present the real values.

Comment 3.10: Page 4 line 188 - correction needed for word 'weekday'.

Answer: Weekday means days of the week without Saturday and Sunday.

Comment 3.11: Page 4 line 267 - it must be defined the expression used for calculation of COD conversion at the experimental part.

Answer: The COD conversion equation is the same as for other types of conversion equation. It was thus not included in the manuscript.

Comment 3.12: Page 9 line 312 - the title of Figure 2 must be a little bit modified. Separate explanation for (A) and (B) which will follow after '.' and before these instead of 'Variation' can be used a formulation as 'Balance of main compounds produced and consumed. (A)...' or an other introductive formulation.

Answer: The title of Figure 2 was changed to take into consideration this comment and be more similar to the other figure titles.

Comment 3.13: Page 9 line 321 - text correction as 'Different operating parameters...'.

Answer: The proposed change has been made.

Comment 3.14: Page 10 line 372 - text correction as '...for the first four phases,...'. (not '5' but 'five').

Answer: The number 4 was changed for the word “four”. However, it is intended that four is written in the text and not five.

Comment 3.15: Page 11 line 380 - Figure 4 - title correction as '....tCOD and sCOD...(A) ....(B) of the effluent in the CSTR ...'

Answer: The proposed change has been made.

Comment 3.16: Page 12 - correction required for:

(i) line 414 - delete text 'that will be converted' and the text will remain as 'but at the cost of globally converting only 70-75% of the tCOD in a CSTR.';

Answer: The proposed change has not been implemented since this formulation was chosen to make sure that it is clear.

(ii) line 421 - correction as 'tCOD in the CSTR is achieved.' (too much 'of');

Answer: The sentence was modified.

(iii) line 438 - possible correction of the following text at the fifth word or related to it and the second word -  'Potential im-provements could further improve the techno-economic viability of such process,...'.

Answer: The word “improve” was changed for “increase”.

Comment 3.17: The authors' contribution must be mentioned.

Answer: The authors' contribution has been added.

Comment 3.18: An English spelling check can be performed for verifying no other mistakes presence.

Answer: The English of the entire manuscript was double-checked.